

# Seed survival of Australian *Acacia* in the Western Cape of South Africa in the presence of biological control agents and given environmental variation

Matthys Strydom[1,2,3], Ruan Veldtman[1,4], Mzabalazo Z. Ngwenya[5,6] and Karen J. Esler[1,2]

[1] Department of Conservation Ecology and Entomology, Stellenbosch University, Matieland, South Africa
[2] Centre of Excellence for Invasion Biology, Stellenbosch University, Matieland, South Africa
[3] Academy for Environmental Leadership SA, Upington, South Africa
[4] South African National Biodiversity Institute, Kirstenbosch National Botanical Garden, Cape Town, South Africa
[5] Statistics in Ecology, Environment and Conservation (SEEC), Department of Statistical Sciences, University of Cape Town, Rondebosch, South Africa
[6] Biometry, Agricultural Research Council, Stellenbosch, South Africa

Corresponding author
Matthys Strydom, matthys@afel.ac, strydomm1987@gmail.com

## ABSTRACT

Studies of invasive Australian *Acacia* have shown that many seeds are still produced and accumulate in soil stored seed banks regardless of the presence of seed-targeting biological control agents. This is despite claims of biological control success, although there is generally a lack of data on the seed production of invasive Australian *Acacia* before and after the release of the respective agents. We aimed to quantify seed production and seed survival of invasive Australian *Acacia* currently under biological control. The seed production and survival (proportion of aborted, predated and surviving seeds) of *A. longifolia*, *A. pycnantha* and *A. saligna* were each studied at four to five sites in the Western Cape of South Africa. The relationships between seed production and stand characteristics were determined and the relative effects of seed predation and abortion on seed survival were established. The investigated invasive Australian *Acacia* produced many seeds that survived the pre-dispersal stage despite long-term presence of released biological control agents. It was shown that seed crop size is the only significant factor influencing seed survival of the studied Australian *Acacia* species. Furthermore, the seeds surviving per tree and per square meter were related to tree size. No quantitative evidence was found to suggest that seed-reducing biological control agents are having an impact on the population dynamics of their Australian *Acacia* hosts. This study illustrates the importance of studying the seed ecology of invasive plants before biological control agents are selected and released.

## INTRODUCTION

Assessment of the seed bank dynamics of invasive alien plants provides a framework to determine when control measures would be most effective during population development (e.g., at what stage are seed banks the smallest) to reach management goals (*Richardson & Kluge, 2008*). In addition, it gives an indication to whether control measurements will be successful before they are applied (*Kriticos et al., 1999*). However, this aspect of the population dynamics of invasive alien plants is generally poorly understood (*Gioria, Pyšek & Moravcová, 2012*). Invasive Australian *Acacia* have become a model system to illustrate the importance of a focus on seed bank dynamics to understand the invasion process as well as to recommend appropriate management interventions (*Richardson & Kluge, 2008*).

Australian *Acacia* are of great commercial value and have been introduced worldwide (*Griffin et al., 2011*). Many of these plants have become naturalised and a few have become invasive (*Richardson et al., 2011*). Invasive Australian *Acacia* often cover vast areas of natural and agricultural land (*Bar Kutiel, Cohen & Shoshany, 2004*; *Henderson, 2007*; *Marchante, Freitas & Hoffmann, 2011b*) and have significant impacts on natural capital (e.g., reduced water availability) (*Le Maitre et al., 2011*; *Rascher et al., 2011*; *Cohen & Bar Kutiel, 2017*). Their impact has led to chemical and mechanical clearing operations (*Van Wilgen et al., 2012*) and biological control programs (*Impson et al., 2011*; *Marchante, Freitas & Hoffmann, 2011a*). Especially, in South Africa many resources have been spent in an effort to control these plants (*Van Wilgen et al., 2012*).

The invasive potential of Australian *Acacia* has been attributed to their pre-adaptation to environmental conditions (*Roux & Middlemiss, 1963*), high growth rates (*Witkowski, 1991*) and copious seed production (*Milton & Hall, 1981*; *Dennill, 1985*; *Neser, 1985*; *Pieterse & Cairns, 1986b*; *Holmes, Dennill & Moll, 1987*; *Donnelly, 1992*; *Impson, Moran & Hoffmann, 2004*). The prolific production of seeds by invasive Australian *Acacia* has been widely proposed and ascribed to a lack of natural enemies (*Van den Berg, 1977*; *Milton, 1980*; *Impson, Moran & Hoffmann, 2004*; *Richardson & Kluge, 2008*); however, the natural enemy release hypothesis remains controversial and lacks quantitative proof (*Maron & Vilà, 2001*; *Colautti et al., 2004*; *Parker, Burkepile & Hay, 2006*).

In contrast to the view that natural enemies are lacking, many studies have demonstrated that Australian *Acacia* seeds may be lost to indigenous insects (*Holmes & Rebelo, 1988*; *Donnelly & Stewart, 1990*; *Pieterse, 1998*), birds (*Middlemiss, 1963*; *Winterbottom, 1970*; *Glyphis, Milton & Siegfried, 1981*; *Pieterse, 1986*) and mammals (*Middlemiss, 1963*; *Holmes, 1990*; *Pieterse & Cairns, 1990*; *Mokotjomela & Hoffmann, 2013*). *Pieterse (1998)* showed that whole seed crops of *A. implexa* may be lost to a native insect. Likewise, *Holmes (1990)* demonstrated that native rodents have the potential to completely consume annual seed crops of invasive Australian *Acacia*. Indigenous seed predators can therefore have substantial impacts on the seed survival of invasive Australian *Acacia*. Nevertheless, in South Africa, the enemy release hypothesis is still used as a basis for the rationale to search for and release biological control agents on Australian *Acacia* (*Van den Berg, 1977*; *Richardson & Kluge, 2008*; *Zachariades et al., 2017*).

Based on this premise, ten biological control agents, namely five seed-feeding weevils (*Melanterius* spp), two bud-galling wasps (*Trichilogaster* spp.), two flower-galling flies (*Dasineura* spp.) and a rust fungus (*Uromycladium* sp.), have been released on Australian *Acacia* in South Africa to decrease both the invasive potential of these plants as well as management costs (*Zimmermann, Moran & Hoffmann, 2004*; *Impson et al., 2011*). The use of seed-reducing agents was further motivated, as these are generally host-specific and would not harm the useful parts of the plants, resolving conflicts of interest for commercially valuable species (e.g., *A. mearnsii* that is used for tannin and wood chip production in South Africa) (*Dennill & Donnelly, 1991*; *Pieterse & Boucher, 1997*; *Impson et al., 2011*; *Zachariades et al., 2017*). All of the agents reduce seed production and usually a combination of two agents, a gall-former and a seed-feeding weevil, is used on a targeted invasive Australian *Acacia* species. The impact of most of these agents has been described as extensive (almost no seeds surviving) or considerable (>50% reduction) (*Impson et al., 2011*; *Zachariades et al., 2017*). It has also been indicated that no (e.g., *A. longifolia*) active management (mechanical clearing) or substantially less management than in the past (e.g., *A. saligna*) is required to control the targeted Australian *Acacia* (*Zimmermann, Moran & Hoffmann, 2004*; *Zachariades et al., 2017*). Based on this apparent success, it is upheld that seed-reducing agents are the only cost-effective and sustainable way to manage these plants (*Richardson & Kluge, 2008*; *Wilson et al., 2011*).

Despite the reported success of using seed-reducing agents to control Australian *Acacia* in South Africa, many agents failed as biological control agents (*Myers, Risley & Eng, 1988*; *Myers & Risley, 2000*). Invasive *Acacia* possess many of the characteristics described as likely candidates for failed use of insect herbivores (*Crawley, 1989*) and more specifically seed-reducing agents (*Myers & Risley, 2000*; *Van Klinken et al., 2003*). These attributes are (as suggested by *Crawley, 1989*): a long growing period, a tough woody stem, high powers of regrowth, high tannin content and large seed banks with protracted dormancy. These are the same qualities possessed by *Vachellia nilotica* (syn *Acacia nilotica*), against which a seed-reducing agent was documented as unsuccessful in Australia (*Kriticos et al., 1999*). Characteristics that make invasive Australian *Acacia* poor candidates for biological control through the use of seed-reducing agents are (as suggested by *Van Klinken et al., 2003*): high plant fecundity, short maturation period, high seed viability and protracted seed dormancy. *Crawley (1989)* and *Kriticos et al. (1999)* further suggest that seed-reducing agents will fail against plants that are not seed limited, i.e., where seeds do not limit recruitment rates and stand density. *Andersen (1989)*, for example, showed that a reduction of 95% in the seed production of four long-lived perennial plants did not impact on their population recruitment.

Despite the claimed impact of released biological control agents, Australian *Acacia* seed banks remain a challenge to management (*Strydom, Esler & Wood, 2012*; *Strydom et al., 2017*). Many Australian *Acacia* are adapted to fire driven systems through the accumulation of persistent seed banks (i.e., storage effects) (*Auld, 1996*). These soil reserves increase in size with time since fire (*Milton & Hall, 1981*). Seed persistence in the soil is the consequence of physically imposed dormancy, requiring hot temperatures (fire or non-fire) or mechanical injury to stimulate germination (*Pieterse & Cairns, 1986a*; *Pieterse & Cairns, 1986b*; *Jeffery,*
*Holmes & Rebelo, 1988*). Recent seed bank studies have shown that Australian *Acacia* seed banks in their invaded ranges in South Africa are still large (1017 to 17,261 seeds m$^{-2}$), currently accumulating (*Strydom, 2012*; *Strydom, Esler & Wood, 2012*; *Strydom et al., 2017*) and are of similar size to that measured prior to biological control agent release (*Strydom, 2012*; *Strydom et al., 2017*). Therefore, these data suggest that seed production and survival are still high and that the released biological control agents are having little overall effect. Consequently, the seed bank data and prior assessments of biological control agent success are contradictory (*Zimmermann, Moran & Hoffmann, 2004*; *Impson et al., 2011*; *Zachariades et al., 2017*). Furthermore, data on the reproductive capacity of invasive Australian *Acacia* are generally scarce (*Gibson et al., 2011*), so making sense of the estimated proportion of seeds lost to biological control is difficult. For example, in South Africa, no seed bank and seed rain data have previously been collected for *A. pycnantha,* or assessed for *A. longifolia* over the past 30 years.

Our study assessed the seeds lost to predation within the pods, as well as the impact of seed predation on the survival of invasive Australian *Acacia* seeds. The seeds surviving per tree (i.e., not lost to predation or abortion during the pre-dispersal phase) as well as the seed rain m$^{-2}$ of *Acacia* cover, i.e., the annual input into the seed bank, were quantified. Rate of seed loss within the soil after a year and the presence of gall-forming agents on trees were also determined. These data therefore allow the assessment of the current impact of the introduced biological control. Therefore, these data were collected to test the following hypothesis that was derived from the results of *Strydom et al. (2017)*:

Seed banks of invasive Australian *Acacia* accumulate within their invaded ranges over time as a consequence of constant seed input in the presence of released gall-forming biological control agents.

## METHODS

### Study sites

Seed survival of *A. longifolia, A. mearnsii, A. pycnantha* and *A. saligna* were studied. Three to five study sites within the Western Cape of South Africa were chosen for each of the four investigated Australian *Acacia* species. All studied sites contained monospecific stands of the investigated species. Sites were selected on the same basis as described in *Strydom et al. (2017)*. In addition to the criteria listed by *Strydom et al. (2017)*, dense stands were also chosen as this is the predominant condition of these plants in South Africa (*Henderson, 2007*; *Henderson & Wilson, 2017*). Sites were also selected to include different levels of precipitation and therefore over an environmental gradient. Investigations at the studied sites were conducted during 2012 and/or 2013.

The study area is characterised by a Mediterranean climate (*Roura-Pascual et al., 2011*). The mean annual rainfall (MAR) over the sampled range varied between 412 to 955 mm with most of the rain (60 to 81%) being received during the winter months (Table S1). The mean annual temperature varied between 15 to 19 °C over the sampled range, the recorded average minimum temperature of the coldest month was 5 to 10 °C while the average maximum temperature of the warmest month was 27 to 33 °C. The first sampled

season (2012) received less rainfall (MAR = 621 mm) than the second sampled season (2013, MAR = 809 mm).

## Sampling procedure

Sampling of trees or selection of plots within sampled monospecific Australian *Acacia* populations, except if otherwise stated, was conducted randomly according to the following described method: sampling was initiated at a random tree or location within the sampled stand. Afterwards the direction of movement (north, east, south or west) as well as the steps taken between the samples was determined from a random numbers table. After the steps taken were completed, the nearest tree was sampled or sampling plot laid out. This was repeated at every site for each of the studied species.

## Visual vs. actual pod estimates

Pods and galls were visually estimated on seven to fifteen pod bearing trees for each species at each site. The visual estimates ranged between 1 to 2,500 pods/galls per tree. Afterwards, these trees were felled, their pods and galls harvested and each pod and gall counted. These data were used to establish relationships between visual estimated pod/gall loads and actual counted pod/gall loads (Fig. S1). This sampling was done at four sites for *A. saligna* during both sampled seasons. For *A. pycnantha* this sampling was conducted at five and three sites during 2012 and 2013 respectively. This same procedure was repeated for *A. longifolia* at one site, during 2013 when an additional 15 trees were sampled (total = 30 trees) to increase the range in tree size sampled. The average stem diameters at breast height (DBH) of the sampled trees for *A. longifolia*, *A. pycnantha* and *A. saligna* were 22.1 mm (6.1 to 49.7 mm), 31 mm (5.1 to 93 mm) and 23.2 mm (1 to 86 mm) respectively.

## Proportion aborted, predated and surviving seeds per tree

The proportion of seeds lost to abortion and predation was determined at five sites for *A. pycnantha* and *A. saligna* during 2012 and 2013. Assessments were also conducted at five sites for *A. longifolia* and three sites for *A. mearnsii* during 2013. Investigations of seed fate (abortion, predation or surviving) commenced when pods of the studied Australian *Acacia* were mature (*Auld & Myerscough, 1986*) during November/December (*Milton, 1982*). Mature or ripe pods were defined as pods that were hardened and had started to open, but still contained all their seeds.

Thirty pods from 8 to 30 fruiting trees (13.9 trees on average) were sampled at each site (thus representing 65 and 67 trees for *A. saligna* and *A. pycnantha* respectively during 2012 and 71, 66, 75 and 45 trees for *A. saligna*, *A. pycnantha*, *A. longifolia* and *A. mearnsii* respectively during 2013). Trees were divided into a top, middle and lower section based on tree size. Within each section ten pods were sampled indiscriminately (*Impson, Moran & Hoffmann, 2004*). If fewer than ten pods were available within a section, those pods were collected within the other sections. All pods were collected if a tree had fewer than 30 pods. Pods were opened, the seeds per pod counted and each seed classified into one of the following three categories: aborted (under-developed and shrivelled seeds), predated (seeds with holes or frass remains), and surviving (fully developed seeds, no visual damage and no disintegration when pressed) (*Brown, Enright & Miller, 2003*).

Seeds assessed as aborted and surviving, however, may contain holes, due to damage caused by insects, not visible without the use of a microscope. Therefore, seeds evaluated as aborted and surviving were separately bulked for each species during 2013. Depending on availability, a subsample of 180 to maximum of 3,600 seeds from each bulked seed pool was taken. Each seed was inspected underneath a microscope for insect damage. If aborted seeds had insect damage they were reclassified as predated. These data were used to calculate the proportion of aborted seeds, visually classified as potentially predated, that were in fact predated (see supplementary information for all calculations). Surviving seeds with insect damage were evaluated as potentially predated as they were not necessarily non-viable.

Surviving seeds with insect damage (34 to maximum of 180 seeds) were subjected to germination (viability) experiments (see below). This was repeated for each species. Only seeds that did not germinate were determined to be predated. These data were used to calculate the proportion of surviving seeds, visually classified as potentially predated, that were in fact predated. From these data the proportion of seeds initially assessed as aborted and surviving that were predated was determined. Furthermore, these data were used to accurately calculate the proportion of surviving, aborted and predated seeds for each tree. From this data the average proportion of surviving, aborted and predated seeds for the studied species were calculated for the respective reproductive seasons investigated. The average seeds per pod for each species were also determined.

## Effect of seed abortion and predation on seed survival

The number of pods per tree was visually estimated on the same trees from which the abovementioned 30 pods were harvested. These visual pod estimates were made before the pods were collected, and were done in the same manner as described above. The relationships established between visual pod estimations and actual counted pods were used to estimate the pods for each tree from their visual pod estimates. The pod estimates, average seeds per pod and the proportion of aborted, predated and surviving seeds per tree were used to estimate the total seeds produced, aborted, predated and surviving for each of the sampled trees. These data were used to assess the effect of seed crop size, seed predation and abortion on seed survival.

## Surviving seeds per tree and per square meter
### Surviving seeds per reproductive tree

The seed rain m$^{-2}$ beneath the canopies of individual reproductive trees was determined. During 2012, this was assessed at five sites for *A. pycnantha* and *A. saligna* (10 sites) and during 2013 at four sites each for *A. longifolia*, *A. pycnantha* and *A. saligna* (12 sites).

Three seed traps were placed beneath the canopy of 25 pod bearing trees (75 traps), at each site. Plastic cups (990 ml) with a diameter of 10.5 cm were sunk into the soil at similar distances from the tree trunk, at angles of approximately 120° from each other. The distance of traps from the tree trunk was dependant on tree canopy size. Traps were placed approximately $\frac{3}{4}$ of the distance from the tree trunk towards the edge of its canopy as seed rain size remains constant underneath the canopy of Australian *Acacia* trees (*Marchante, Freitas & Hoffmann, 2010*). Seed traps were placed in position before
seed fall of the investigated species commenced (October/November) and were retrieved after seed fall had stopped (December to March) (*Milton, 1982*). The seeds within each retrieved trap were counted. As there was some overlap between the canopies of the trees, the number of seeds collected in each trap was divided by the average number of fruiting trees within a radius of 1.5 m from the seed traps (see below). Afterwards the average seeds per cup for each tree were calculated. These data were used to calculate the average seed rain m$^{-2}$ beneath the canopy of each tree. In addition, mean canopy diameter (mean of two measurements at right angles to each other) and DBH of the selected trees were measured.

The surviving seeds of each tree (i.e., seeds that were not lost to abortion and predation) were determined by multiplying its estimated canopy area with its estimated seed rain m$^{-2}$. Canopy area (m$^{-2}$) for each tree was calculated using the area formula for a circle ($A = \pi r^2$, $r$ = mean canopy diameter/2). These data were used to establish whether a relationship between the seeds surviving per tree and DBH exists. Such a relationship would serve as an approximation of the relationship between the seeds surviving per tree and time, as stem diameter is a proxy of tree age (*Spooner et al., 2004*). In order to assess the average seeds surviving per tree at different time steps, the stem diameter of each studied species was divided into different size classes and the average seeds surviving per tree for each class calculated.

### *Australian Acacia seed rain m$^{-2}$*

The seed rain m$^{-2}$ within Australian *Acacia* stands was determined at four sites each for *A. longifolia*, *A. pycnantha* and *A. saligna* during 2013 (total of 12 sites). To assess the seed rain (m$^{-2}$) thirty seed traps, as described above, were randomly sunk into the soil throughout the investigated Australian *Acacia* stands. Traps were placed out and retrieved during the same time as stated for the previous experiment. After retrieval, the seeds in each trap were counted. At each sampling point, the trees within a circle (1.5 m radius), with the seed trap as midpoint were counted (*Strydom et al., 2017*). If the canopy of trees not inside the circle crossed the seed trap, these trees were also counted (*Strydom et al., 2017*). It was also noted whether each counted tree had pods. The DBH of all trees whose canopies intersected the sampling point was measured, because the largest proportion of Australian *Acacia* seed fall directly beneath their canopy (*Milton & Hall, 1981*; *Marchante, Freitas & Hoffmann, 2010*). It was also noted whether the stems of these trees fell within or outside the circle. The DBH of up to three additional trees within the circle was measured if the DBH of all the trees in the circle was not already measured (*Strydom et al., 2017*). These data were used to calculate tree density, the proportion of fruiting trees, average fruiting trees within 1.5 m of seed traps and average stem diameter. Furthermore, it was used to determine whether relationships between tree density and the seed rain m$^{-2}$ or between tree size and the seed rain m$^{-2}$ exist. In order to approximate the seed rain m$^{-2}$ over time, the average stem diameter of each studied species was divided into different size classes and the average seed rain m$^{-2}$ for each class calculated.

### Seed viability

A sample of 100 seeds assessed as surviving was taken from the seed pool of each species at each site. The seed coats of these seeds were chipped at the distal end and 25 placed in each

of four petri-dishes containing two filter paper discs moistened with 10 ml water (*Pieterse & Cairns, 1986a*). Petri-dishes were placed in plastic bags to prevent moisture loss (*Pieterse & Cairns, 1986a*) and incubated at 25 °C in the dark (*Hendry & Van Staden, 1982*). After 3 days, seeds were checked for germination and thereafter daily for two weeks. Germination was assumed if the radicle was at least 1 mm long (*Holmes, 1988*).

## Seed Germination/Decay

Seven hundred and fifty seeds, from the surviving seed pool (seeds were bulked for each site) of *A. pycnantha* and *A. saligna* at three sites, were selected for seed burial trials during 2012. The same quantity of seeds were collected and used for *A. mearnsii* and *A. longifolia* at three and one site respectively. Shade net (75%) was used to make 5 × 5 cm packets in which 30 seeds were placed (25 packets per site). The packets were buried five centimetres deep and one meter apart, along five transects (one meter apart), at each site where the seeds were collected. The packets were buried during June 2013 and left for a year, after which they were retrieved. The collected packets were opened and the remaining seeds in each packet counted. Furthermore, initial dormancy of seeds at the collected sites was also determined before burial. To assess this, an additional one hundred seeds were selected from the surviving seed pool of *A. pycnantha* and *A. saligna*, while the same quantity of seeds was collected at the *A. mearnsii* and *A. longifolia* sites. The same procedure as described above for viability testing was followed, except seeds were not chipped at their distal ends. The seeds that germinated were used to calculate the proportion of initially non-dormant seeds.

## Biological control and gall-forming agent presence

The gall-forming biological control agents of the studied species were released 25 to 30 years ago and are present wherever their host plants occur within South Africa (*Olckers & Hill, 1999*; *Dorchin, Cramer & Hoffmann, 2006*; *Impson et al., 2011*). Since their establishment across their hosts' distribution range in South Africa, sufficient time has passed for these agents to have visual and quantifiable impact on the populations of their hosts (*Dennill, 1985*; *Dennill, 1987*; *Morris, 1997*; *Hoffmann et al., 2002*; *Wood & Morris, 2007*). In order to assess the impact of these agents on seed rain and therefore seed survival over time, a chronosequence was established. This was achieved through using stem diameter as a proxy of tree age (*Spooner et al., 2004*). The relationship between average seed rain m$^{-2}$ and stem diameter and therefore time was determined through the use of quantile regression (see below). To estimate the average seed survival of the studied species under the average impact of their gall-forming biological control agents over time, average seed rain (m$^{-2}$) for tree size classes was established. This was achieved by dividing the trees into different tree size classes based on DBH and then calculating the average seed rain m$^{-2}$ for each tree size class. The average seed rain m$^{-2}$ for each size class or time step also includes the influence of other limiting factors (e.g., water availability, nutrient availability, competition etc.). Furthermore, this measurement also takes into account the average time taken for agents to establish (locate, survive and reproduce) within populations as well as their average annual accrual rates after disturbance events (*Strydom et al., 2017*). This measurement therefore

**Table 1** Maximal models (models containing all explanatory variables) used in analyses.

| Response variable | Explanatory variables | Random Effects |
|---|---|---|
| Seeds surviving tree$^{-1}$ | Seed crop size of tree + seeds aborted per tree + seeds predated per tree + species | Site and Year |
| Seeds surviving tree$^{-1}$ | Stem diameter + species | Site and Year |
| Seeds surviving m$^{-2}$ | Average stem diameter + tree density + species | Site |

represents an estimate of seed survival over time under the established dynamics between the gall-forming biological control agents and their invasive Australian *Acacia* hosts. This same methodology has been used to determine the impact of gall-forming biological control agents on the seed banks of invasive Australian *Acacia* over time (*Strydom et al., 2017*).

At the studied sites, the presence of gall-forming biological control agents was also determined. If gall structures were located on trees within the investigated populations, the gall-forming control agents of the studied species were assessed as present (*Strydom et al., 2017*).

## Statistical analyses

Generalised linear mixed models (GLMMs) were fitted to the seed fate (surviving, predated and aborted seeds per tree) data to establish whether seed predation and abortion significantly affects seed survival. GLMMs were also used to determine whether the surviving seeds produced per tree (SST) and the seed rain m$^{-2}$ were significantly related to tree and stand characteristics respectively. Quantile regressions were then employed to assess the relationship between the SST and stem diameter and seed rain m$^{-2}$ and average stem diameter. All statistical analyses were conducted in R 3.1.3 (*R Core Team, 2015*).

### Generalised linear mixed models

GLMMs were used as the response variables in all the fitted models were count data and random effects were present (*Bolker et al., 2009*). Models with a log link function and a negative binomial error distribution were fitted to the data. A negative binomial error distribution was used to account for overdispersion (*Grueber et al., 2011*). The explanatory variables were standardised to remove the effect of scale on the parameter estimates (*Grueber et al., 2011*). Furthermore, maximum likelihood estimation was used to obtain model parameters (*Johnson & Omland, 2004*).

After assembling the maximal model (model containing all explanatory variables) (see Table 1), a submodel set was generated from which a 95% confidence model set was obtained (*Grueber et al., 2011*). Models were compared through the use of information theoretic (I-T) model procedures based on Akaike's information criterion (AIC) (*Burnham, Anderson & Huyvaert, 2011*). The model second order AIC value, the AICc, was used to select the best model. If the best model had a corrected Akaike weight (AICcWt) of less than 0.9, model averaging was considered. Model averaging was conducted on the 95% model confidence set to obtain parameter estimates, parameter confidence intervals and relative parameter importance (*Grueber et al., 2011*).

*Quantile regression*
The set of five hierarchical models proposed by *Huisman, Olff & Fresco (1993)* was used to determine the response curve of the SST and seed rain $m^{-2}$ to stem diameter and average stem diameter respectively over the sampled environmental gradient. This was done for *A. longifolia*, *A. pycnantha* and *A. saligna*. These models were used as they are appropriate to discover non-linear response curves along environmental gradients (*Oksanen & Minchin, 2002*).

The highest order model, from which four lower order models can be derived, is defined by *Huisman, Olff & Fresco (1993)* as follows:

$$y = M \frac{1}{1+e^{a+bx}} \frac{1}{1+e^{c-dx}}$$

where $y$ is the response variable, $x$ the explanatory variable, $a$, $b$, $c$ and $d$ are the parameters to be approximated and $M$ a constant, which refers to the maximal attainable value for y. The value of $M$ was set by determining the highest recorded number of seeds per tree or seed rain $m^{-2}$ of each species. All five models were fitted to the data using quantile regression models at the 95th percentile. Afterwards the best fitting model was selected according to the AIC value. These procedures were done in the same manner as described in *Strydom et al. (2017)*.

## RESULTS

### Gall-forming biological control agent presence
The gall agents of *A. longifolia* (*Trichilogaster acaciaelongifoliae*), *A. pycnantha* (*T. signiventris*) and *A. saligna* (*Uromycladium tepperianum*) were present at all the sampled sites. However, for *A. mearnsii*, its gall agent (*Dasineura rubiformis*) was only present at one of the three sampled sites, viz. Rivendale.

### Surviving seeds per tree
Many surviving seeds (i.e., not aborted or predated) were produced by the reproductive trees of the studied Australian *Acacia* (Table 2) irrespective of the seeds lost to predation and abortion (Table 3). This was confirmed by the seeds surviving per tree not being significantly related to seed predation or abortion in the fitted GLMMs (Tables 4 and 5). However, the seeds surviving per tree were significantly and positively related to seed crop size (Tables 4 and 5) and were also significantly positively related to stem diameter (Table 6). Therefore, the surviving seeds per tree increased with tree size and by inference, with tree age (Figs. 1A–1F). This relationship indicates that more seeds survive as seed crop size increases. The highest numbers of recorded surviving seeds produced by a tree were 35,865, 17,729 and 14,007 seeds for *A. longifolia*, *A. pycnantha* and *A. saligna* respectively. The viability of the surviving seeds of all the studied species was high. On average the seed viability of *A. longifolia*, *A. mearnsii*, *A. pycnantha* and *A. saligna* was 97.1% (94.7 to 100%), 98.5% (97 to 99%), 96.6% (93.4 to 98.5%) and 97.8% (95 to 99%) respectively.

### Seed rain
The abundance of seeds produced by reproductive trees led to many seeds surviving per $m^{-2}$ as shown by the seed rain of the studied Australian *Acacia* (Figs. 2A–2F). The seed rain

**Table 2** Average surviving seeds per reproductive tree (±SE) as well as the seed rain m$^{-2}$ (±SE) of three Australian *Acacia* during 2012 and 2013 in the Western Cape of South Africa.

| Species | Surviving seeds per tree | | Seeds rain (m$^{-2}$) |
|---|---|---|---|
| | 2012 | 2013 | 2013 |
| *A. saligna* | 1,454 (190) | 1,574 (229) | 1,942 (153) |
| *A. pycnantha* | 460 (96) | 107 (20) | 307 (45) |
| *A. longifolia* | nm | 2,154 (439) | 1,006 (120) |

Notes.
nm, not measured.

**Table 3** Proportion of aborted, predated and surviving seeds per reproductive tree of four Australian *Acacia* during 2012 and 2013 in the Western Cape of South Africa. Values in parenthesis indicate the standard error.

| Species | % Aborted | | % Predated | | % Surviving | |
|---|---|---|---|---|---|---|
| | 2012 | 2013 | 2012 | 2013 | 2012 | 2013 |
| *A. saligna* | 29.6 (1.9) | 23.6 (1.7) | 5.6 (1.5) | 16.0 (2.9) | 64.8 (2.6) | 60.4 (3.6) |
| *A. pycnantha* | 22.7 (1.4) | 20.4 (1.3) | 24.6 (1.8) | 22.6 (1.6) | 52.7 (2.7) | 57.0 (2.0) |
| *A. longifolia* | nm | 21.3 (1.2) | nm | 39.1 (1.9) | nm | 39.6 (2.0) |
| *A. mearnsii* | nm | 30.6 (3.2) | nm | 41.9 (2.2) | nm | 27.5 (2.3) |

Notes.
nm, not measured.

**Table 4** Results of model averaging over the fitted models with surviving seeds per tree as response variables for each studied species. The effect of each parameter on the response variable is shown.

| Parameters | Estimate | Standard error | p-value | Confidence interval | Relative importance |
|---|---|---|---|---|---|
| Intercept | 7.53 | 0.24 | <0.001 | (7.05, 8.01) | |
| Crop size | 1.65 | 0.27 | <0.001 | (1.12, 2.17) | 1 |
| Seeds predated | −0.07 | 0.15 | 0.669 | (−0.64, 0.20) | 0.30 |
| Seeds aborted | −0.03 | 0.13 | 0.830 | (−0.58, 0.36) | 0.25 |
| *A. pycnantha* | 0.01 | 0.15 | 0.922 | (−0.82, 1.27) | 0.06 |
| *A. saligna* | 0.02 | 0.16 | 0.915 | (−0.85, 1.37) | 0.06 |

m$^{-2}$ of the investigated Australian *Acacia* was found to be significantly related to average stem diameter (Tables 7 and 8). Across all species, the seed rain m$^{-2}$ increased with stem diameter until a maximal point was reached after which the seed rain m$^{-2}$ decreased with a further increase in stem diameter (Figs. 2A–2C). The average seed rain m$^{-2}$, for stem diameter classes of the studied species (Figs. 2D–2F), generally followed the same trend of increase and decline with an increase in stem diameter class size as described above for their associated quantile regressions (Figs. 2A–2C). The seed rain m$^{-2}$ of *A. pycnantha*, *A. saligna*, and *A. longifolia* is at a maximum when average stem diameter for these species are 39 mm, 44 mm and 171 mm (Figs. 2A–2C). At the maximum point with stem diameter as the explanatory variable, the seed rain m$^{-2}$ covered by *A. pycnantha*, *A. saligna* and *A. longifolia* was estimated to be 1,477 seeds m$^{-2}$, 6,095 seeds m$^{-2}$, and 4,800 seeds m$^{-2}$ respectively.

**Table 5 Best candidate models (95% model confidence set) predicting the surviving seeds per tree.** Model AICc values, model weights ($\omega_i$), cumulative model weights (acc $\omega_i$) and Laplace approximations (LL) is shown. Model averaging was conducted over the candidate models to obtain parameter estimates, parameter confidence intervals and relative parameter importance. Model averaging results shown in Table 4.

| Model | k | AICc | $\Delta_i$ | $\omega_i$ | Acc $\omega_i$ | LL |
|---|---|---|---|---|---|---|
| Crop size | 6 | 6376.99 | 0.00 | 0.46 | 0.46 | −3182.37 |
| Crop size + seeds predated | 7 | 6378.49 | 1.50 | 0.22 | 0.68 | −3182.08 |
| Crop size + seeds aborted | 7 | 6378.98 | 2.00 | 0.17 | 0.85 | −3182.33 |
| Crop size + seeds predated + seeds aborted | 8 | 6380.40 | 3.41 | 0.08 | 0.93 | −3181.99 |
| Crop size + species | 8 | 6380.93 | 3.49 | 0.06 | 0.99 | −3182.26 |

**Table 6 Response of viable seeds per tree to stem diameter.** Data analysed with a generalized linear mixed model fit by the Laplace approximation with a negative binomial error distribution.

| Fixed effect | Estimate ± SE | $t$-value | $p$-value |
|---|---|---|---|
| Intercept | 6.44 ± 0.82 | 7.89 | <0.001 |
| Stem diameter | 0.92 ± 0.09 | 10.35 | <0.001 |
| *A. pycnantha* | 0.09 ± 0.91 | 0.10 | 0.918 |
| *A. saligna* | 0.63 ± 0.87 | 0.72 | 0.470 |

**Notes.**
Model AIC value, 7682.7; Model deviance, 7666.7; Random effects, Site; StDev, 0.53; Year, StDev, 0.95; Site factor, 14.

## Proportion of seeds persisting for one year or more

Seeds were retrieved successfully at nine of the ten sites as one of the *A. mearnsii* sites was mechanically cleared during the duration of the experiment. *Acacia longifolia* lost the highest proportion of its produced seeds after a year's burial (78%), followed by *A. mearnsii* (77%), *A. saligna* (65%) and finally *A. pycnantha* (58%) (Table S2).

## DISCUSSION

Despite the presence of seed-reducing agents in the pre-dispersal phase, seed production and seed survival of Australian *Acacia* remain high, confirming the patterns described for seed bank accumulation by *Strydom et al. (2017)*. The annual seed production and seed banks are therefore larger than required to re-establish seedling densities as well as mature stand densities after disturbance events (*Strydom et al., 2017*). Therefore, it is argued that the population dynamics of the investigated Australian *Acacia* are likely unaffected by the released seed-reducing biological control agents in South Africa.

The current study confirmed the prediction of seed bank studies (*Strydom, 2012*; *Strydom et al., 2017*), that many seeds are still produced, survive (despite the presences of seed-reducing biological control agents) and are constantly lost (to either germination or decay) from the seed bank. Many seeds, not lost to seed abortion or predation, were produced by individual reproductive trees of all the studied Australian *Acacia*. This translated into the investigated Australian *Acacia* producing many seeds m$^{-2}$. However, a large proportion of these seeds is lost to either germination or decay within the first year. This illustrates that continual seed production is required to maintain the seed banks of

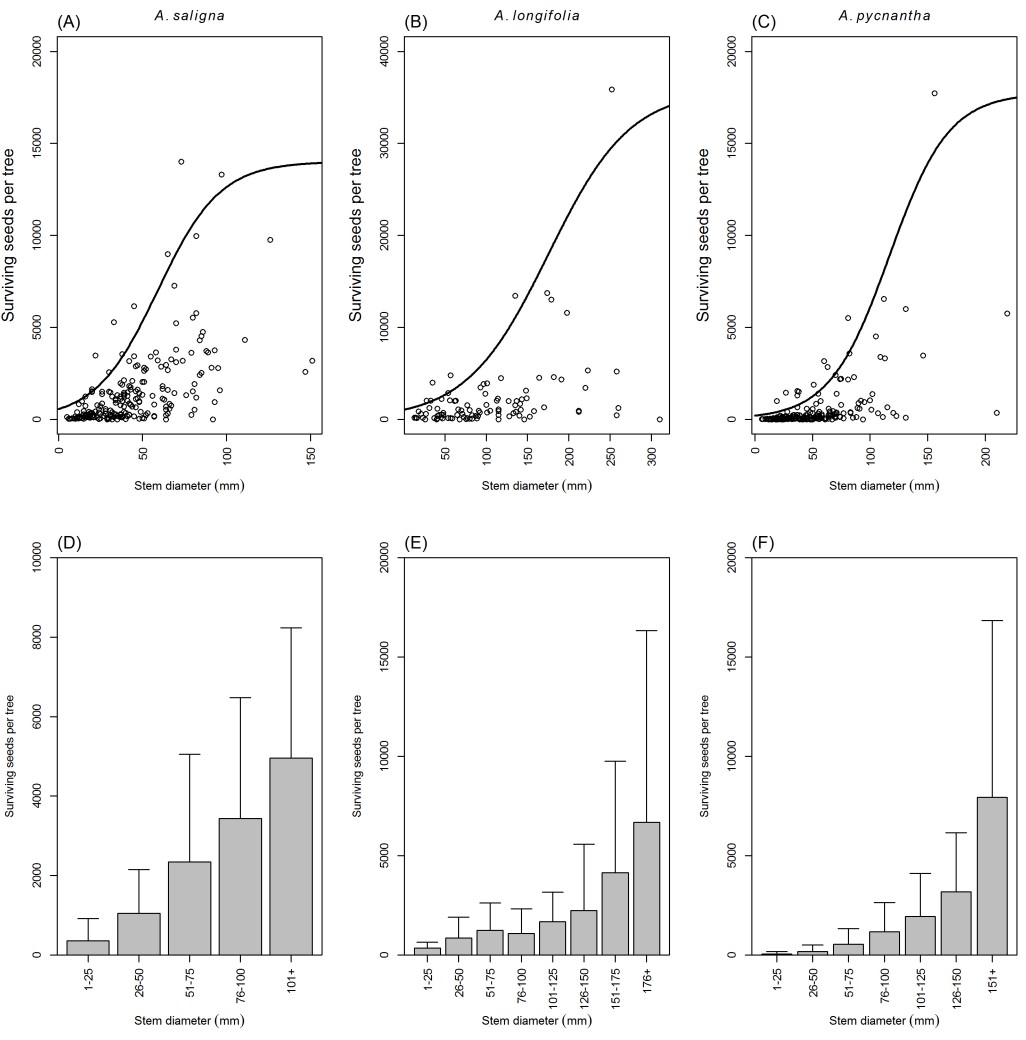

**Figure 1** **Australian *Acacia* seed survival for reproductive trees relative to tree size.** (A–C) The response of seeds surviving per reproductive tree to stem diameter, estimated by 95 % quantile regressions, of *A. saligna*, *A. longifolia* and *A. pycnantha*. (D–F) The average seeds surviving per reproductive tree (+SD) for stem diameter classes of three Australian *Acacia* in the Western Cape of South Africa.

these species and that current seed banks of the investigated species are not the consequence of seed input before the release of their associated gall-forming biological control agents. This conclusion is further supported by the relationship between the seed rain and stem diameter mirroring the relationship between the seed bank and stem diameter of the investigated Australian *Acacia* as established by *Strydom et al. (2017)*. Based on the seed bank (*Strydom et al., 2017*) and seed rain developmental curves, management programs should focus on removing populations during the seedling phase. This will prevent the accumulation of large seed banks and will ensure the management of fewer individuals over time (*Strydom et al., 2017*).

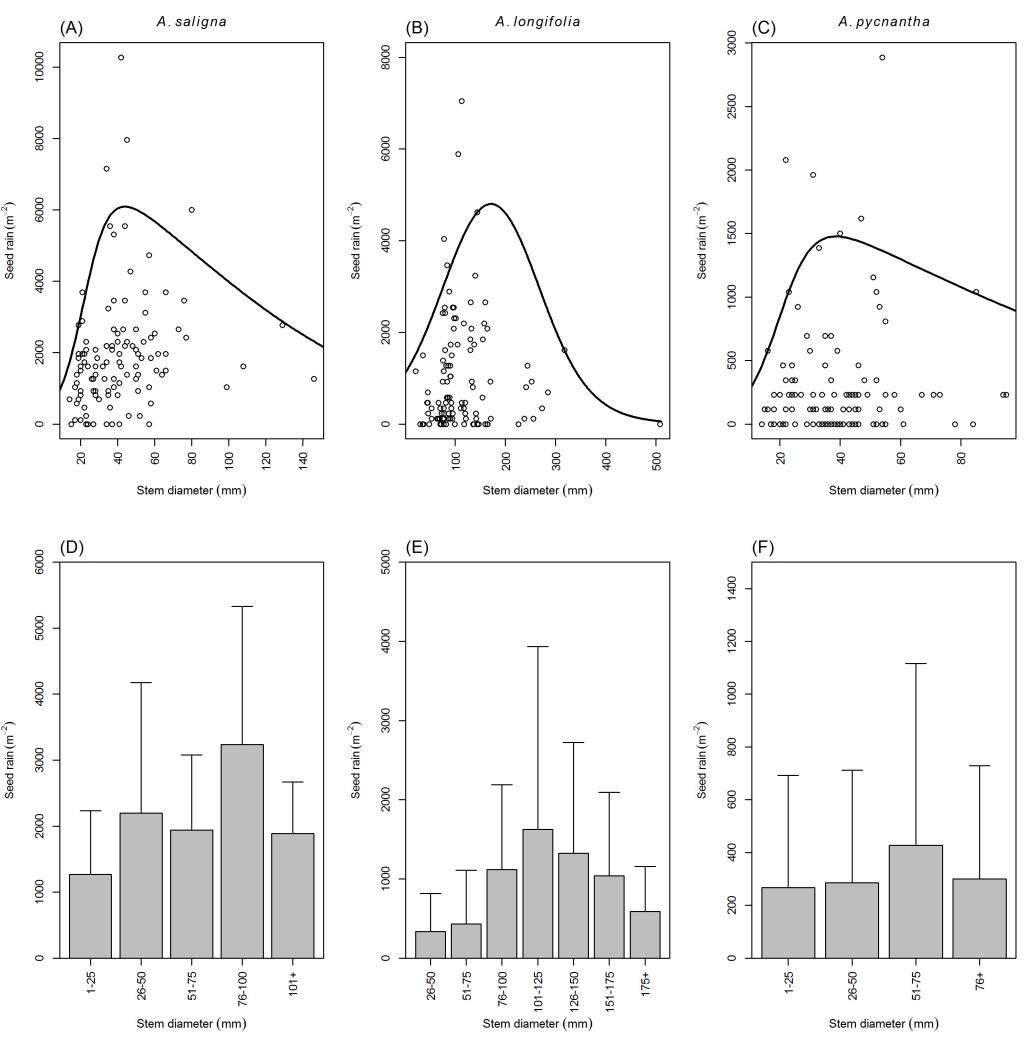

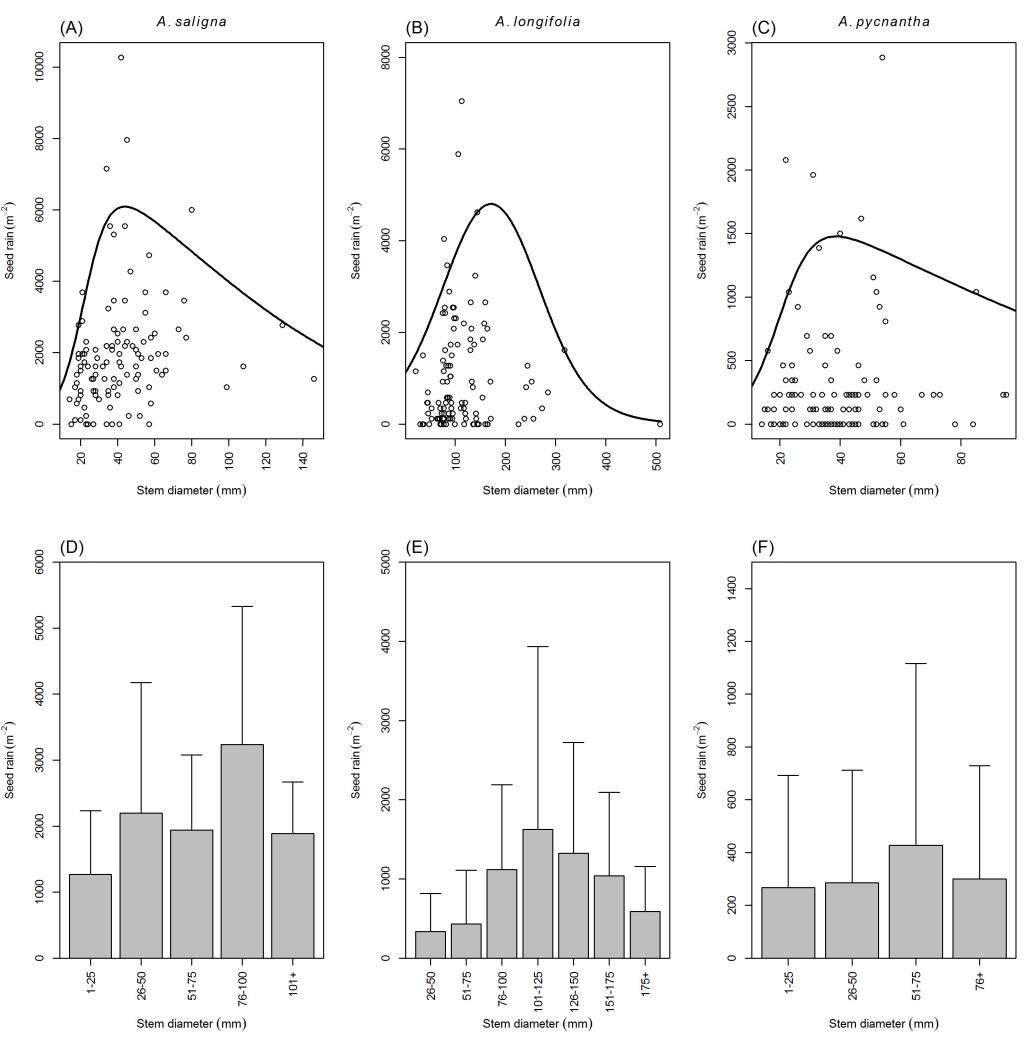

**Figure 2  Australian *Acacia* seed rain relative to average tree size.** (A-C) The response of seed rain m$^{-2}$ to average stem diameter, estimated by 95 % quantile regressions, of *A. saligna, A. longifolia* and *A. pycnantha*. (D-F) The average seed rain m$^{-2}$ (+SD) for stem diameter classes of three Australian *Acacia* in the Western Cape of South Africa.

**Table 7  Results of model averaging over the fitted models with seed rain m$^{-2}$ as response variables for each studied species.** The effect of each parameter on the response variable is shown.

| Parameters | Estimate | Standard error | *p*-value | Confidence interval | Relative importance |
|---|---|---|---|---|---|
| Intercept | 6.27 | 0.38 | <0.001 | (5.52, 7.02) | |
| Average stem diameter | 0.35 | 0.12 | 0.003 | (0.12, 0.57) | 1 |
| Tree density | 0.01 | 0.05 | 0.831 | (−0.14, 0.22) | 0.28 |
| *A. pycnantha* | −0.62 | 0.54 | 0.253 | (−1.68, 0.44) | 1 |
| *A. saligna* | 1.42 | 0.54 | 0.009 | (0.36, 2.47) | 1 |

**Table 8  Best candidate models (95% model confidence set) predicting seed rain m$^{-2}$.** Model AICc values, model weights ($\omega_i$), cumulative model weights (acc $\omega_i$) and Laplace approximations (LL) is shown. Model averaging was conducted over the candidate models to obtain parameter estimates, parameter confidence intervals and relative parameter importance. Model averaging results shown in Table 7.

| Model | k | AICc | $\Delta_i$ | $\omega_i$ | Acc $\omega_i$ | LL |
|---|---|---|---|---|---|---|
| Stem diameter + species | 6 | 5006.54 | 0.00 | 0.72 | 0.72 | −2497.14 |
| Stem diameter + tree density + species | 7 | 5008.43 | 1.90 | 0.28 | 1 | −2497.05 |

Stem diameter, and consequently tree age (as stem diameter is a proxy of tree age) (*Spooner et al., 2004*), was an important determinant of reproductive capacity of invasive Australian *Acacia*, both on an individual tree and population level. Therefore, when comparing seed production of individual trees or populations, within and between species, it is important to consider tree size. Therefore, our findings cannot be compared to that of many previous studies on Australian *Acacia* (*Milton & Hall, 1981*; *Weiss, 1983*; *Auld, 1996*; *Pieterse & Cairns, 1988*; *Holmes, 1990*; *Brown, Enright & Miller, 2003*), as tree size or stand age was not reported. The only studied species for which comparative data of this nature are available is *A. saligna* (*Wood & Morris, 2007*; *Strydom, 2012*).

Despite *Wood & Morris (2007)* suggesting the opposite, there is no clear proof that the seed rain m$^{-2}$ of *A. saligna* has declined in the presence of the gall rust fungus, *U. tepperianum*. *Wood & Morris (2007)* estimated seed rain m$^{-2}$ during 1989 and 2004. However, the tree density estimations of 1989 used in their calculations may not have been representative of the investigated sites (tree densities of other sites or from 1991 were used) (*Morris, 1997*; *Wood & Morris, 2007*). Therefore, only the data from 2004 can be used for comparative purposes. Furthermore, the proportion of seeds lost to abortion and predation was not taken into account in their calculations. Consequently, *Wood & Morris (2007)* estimated the seeds produced m$^{-2}$ and not the seeds surviving m$^{-2}$, i.e., the seed rain m$^{-2}$. Despite this, our study's seed rain estimate of *A. saligna* (1,514 seeds m$^{-2}$, stem diameter = 41.5 mm) was higher than the total seed production estimate of *Wood & Morris (2007)* during 2004 (1,429 seeds m$^{-2}$, stem diameter 41.6 mm). Moreover, the seed rain m$^{-2}$ estimate of *A. saligna* for our study is similar to seed production measured in sites of similar type and tree size by *Strydom (2012)*.

The survival of seeds produced by invasive Australian *Acacia* was not significantly influenced by seed predation, with seed crop size being the only significant factor influencing the quantity of surviving seeds. This indicates that the studied species satiate their seed predators. This is expected given that Australian *Acacia* in their native environments have also been shown to satiate seed predators (*Auld, 1996*; *Auld & Myerscough, 1986*), while seed satiation as a defence mechanism has also been shown for other legumes and plant species (*Janzen, 1969*; *Janzen, 1971*; *Kelly & Stork, 2002*). The recorded seed predation levels may be caused by indigenous (*Holmes & Rebelo, 1988*; *Pieterse, 1998*) and/or introduced insect agents (*Impson et al., 2011*). It is therefore concluded that the introduced *Melanterius* spp. of the studied Australian *Acacia* does not have a significant impact on the seed survival within the investigated *Acacia* populations.
This contradicts previous claims about the effectiveness of the released seed-feeding weevils (*Olckers & Hill, 1999*; *Impson et al., 2011*).

Predation estimates in the absence of other population dynamic parameters are misleading and of little value (*Janzen, 1971*; *Hill, Gourlay & Martin, 1991*; *Kriticos et al., 1999*), because high levels of predation may not have significant impact on population dynamics (*Andersen, 1989*). We found this to be the case in our study. This is concerning as previous studies (except *Donnelly & Hoffmann, 2004*) or reviews (*Olckers & Hill, 1999*; *Impson, Moran & Hoffmann, 2004*; *Impson et al., 2011*) expressed weevil damage as percentages without measuring or indicating seed production on either a tree or population level. Furthermore, the percentage damage caused by weevils, as quantified by previous studies (*Donnelly & Hoffmann, 2004*; *Impson, Moran & Hoffmann, 2004*), was expressed without considering seed abortion. This was based on the assumption that aborted seeds have no bearing on the potential seeds that may be lost to seed feeding weevils belonging to the genus *Melanterius* (*Auld & Myerscough, 1986*). However, *Peguero, Bonal & Espelta (2014)* have shown that seed abortion may influence the seeds of invasive *A. pennatula* lost to predation in South America. Moreover, the total seeds lost to predation will be over- or underestimated if seed abortion is not taken into account.

The lack of impact of *Melanterius* spp. on seed survival may be explained by their absence at the studied sites. However, adult weevils were often observed at the studied locations for all species, except within populations of *A. saligna* (personal obs.). Especially for *M. ventralis* this explanation, for its lack of impact on seed production, is unlikely. *Melanterius ventralis* was released during 1985 (*Impson et al., 2011*), and has since established and increased in abundance throughout its host's distribution range (*Impson & Moran, 2003*; *Olckers & Hill, 1999*), and has had its impact assessed as extensive (*Impson et al., 2011*). *Impson et al. (2011)* have stated that there is no doubt that *M. ventralis* has contributed to the overall decline in the extent and abundance of *A. longifolia*. In contrast, the data presented here suggest that *M. ventralis* is having no measurable impact on the seed survival of *A. longifolia*.

Despite the presence of released biological control agents, seed banks remained large and viable (*Strydom et al., 2017*). Before and after the seed rain, seed bank size of invasive Australian *Acacia* (*A. longifolia*, *A. pycnantha* and *A. saligna*) was estimated to be approximately 1,017 to 17,261 seeds $m^{-2}$ and 2023 to 17,569 seeds $m^{-2}$ respectively. Seed banks of similar size have been shown to give rise to seedling densities of 19 to 1,200 seedlings $m^{-2}$ after disturbance events under a wide range of environmental conditions (*Milton & Hall, 1981*; *Pieterse & Cairns, 1986b*; *Holmes, Dennill & Moll, 1987*; *Pieterse & Cairns, 1987*; *Holmes, 1988*; *Jasson, 2005*; *Merrett, 2013*). *Holmes & Cowling (1997)* indicated that as few as 10 Australian *Acacia* seedlings $m^{-2}$ are able to form a closed canopy. These seedling densities will eventually self-thin to form mature stands, with closed canopies, of 0.52 trees $m^{-2}$ (*Milton & Siegfried, 1981*). Therefore, this provides further evidence that the released biological control agents are having no significant impact on the population dynamics of their host plants. Current data suggests that Australian *Acacia* in their invaded ranges in South Africa are not seed limited. Consequently, the conclusions of *Strydom et al. (2017)* are supported by our study.

Historical and current distribution and abundance data of the studied plants confirm the findings of the current investigation. The current distribution ranges (*Henderson, 1998*; *Henderson, 2007*) of investigated plants includes their historic distribution ranges before the release of their associated biological control agents (*Stirton et al., 1978*). Since the release of the biological control agents of the studied plants, the area occupied by these plants has also increased over time (*Henderson & Wilson, 2017*). This increase in extent could only have been the consequence of less dense stands becoming denser or trees establishing in previously unoccupied locations. Both these conditions require the production of seeds as these plants do not reproduce vegetatively (*Stirton et al., 1978*). This shows that despite the impact of the released biological control agents, enough seeds are produced to establish new and replace existing populations. The distribution and abundance data (*Henderson, 1998*; *Henderson, 2007*; *Henderson & Wilson, 2017*) further suggest that the findings of the current investigation in dense monospecific stands are also relevant to less dense stands.

*Moran, Hoffmann & Olckers (2003)*, *Impson, Moran & Hoffmann (2004)* and *Zimmermann, Moran & Hoffmann (2004)* have all stated that the real benefits of the seed reducing agents are realised for management through the apparent reduction in seed numbers, leading to fewer seedlings having to be managed over time, and therefore a decrease in potential management costs. However, there is no quantitative proof that current seed predation levels are leading to decreased seed bank sizes and consequently fewer seedlings. On the contrary, Australian *Acacia* appear not to be seed limited and current seed production and seed banks are several orders of magnitude larger than that required to re-establish Australian *Acacia* populations after disturbance events (e.g., clearing). The increase in the area occupied by these plants has also occurred in the presence of continual removal of fire wood (*Kull et al., 2011*), a national scale clearing program that has been operating for more than a decade (*Van Wilgen et al., 2012*) and land use change. We conclude that the released biological control agents are not exerting any significant control with no apparent benefit for clearing operations.

## CONCLUSION

Seed production and seed survival of invasive Australian *Acacia* in South Africa are still high. Furthermore, based on seed production and seed bank data the released biological control agents are exerting no significant impact on the population dynamics of their Australian *Acacia* hosts. The released seed-reducing agents also have no benefit for clearing operations. The failure of these agents is not surprising as evidence from other studies suggests that Australian *Acacia* which are mass seeders are poor candidates for biological control through the use of seed-reducing insects. This study supports the conclusion by *Kriticos et al. (1999)* that the population dynamics of invasive plants should be conducted before biological control agents are released. Furthermore, tree size is an important predictor of seed production and survival. The seed banks of Australian *Acacia* also mirror their seed production. Consequently, management should be done during the seedling phase to prevent the accumulation of seed banks of these invasive alien plants.

## ACKNOWLEDGEMENTS

All landowners and managers are thanked for permission to work on their land. The ARC-ISCW is acknowledged for weather data. Colin Tucker, Stembiso Gumede and Clare Gordon for help in the laboratory. Dr. G.J. Strydom is also gratefully acknowledged for his assistance in the field.

### Funding

This research was undertaken with financial support from the DST-NRF Centre of Excellence for Invasion Biology and Working for Water Programme through their collaborative research project on "Integrated Management of invasive alien species in South Africa", and in part by the National Research foundation of South Africa (Grant Number 103841 to Prof. Karen Joan Esler). There was no additional external funding received for this study. The funders had no role in study design, data collection and analysis, decision to publish, or preparation of the manuscript.

### Grant Disclosures

The following grant information was disclosed by the authors:
DST-NRF Centre of Excellence for Invasion Biology and Working for Water Programme.
National Research foundation of South Africa: 103841.

### Competing Interests

The authors declare there are no competing interests.

### Author Contributions

- Matthys Strydom conceived and designed the experiments, performed the experiments, analyzed the data, contributed reagents/materials/analysis tools, prepared figures and/or tables, authored or reviewed drafts of the paper, approved the final draft.
- Ruan Veldtman and Karen J. Esler conceived and designed the experiments, contributed reagents/materials/analysis tools, authored or reviewed drafts of the paper, approved the final draft.
- Mzabalazo Z. Ngwenya analyzed the data, contributed reagents/materials/analysis tools, prepared figures and/or tables, authored or reviewed drafts of the paper, approved the final draft.

### Data Availability

   The raw data is available as Data S1.

### Supplemental Information

Supplemental information for this article can be found online at http://dx.doi.org/10.7717/peerj.6816#supplemental-information.

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
