# Peer review of "Seed survival of Australian Acacia in the Western Cape of South Africa in the presence of biological control agents and given environmental variation"

_PeerJ, doi:10.7717/peerj.6816_

## Round 0.1 · original submission · Major Revisions

Reviewer #1 identified a series of sentences in the Methods section that are directly copied from Strydom et al., 2017. Even though Strydom et al., 2017 might be seen as a “companion paper”, direct copying of sentences from your other publication (“self plagiarism”) must always be avoided. I cursorily checked other sections of the manuscript and I found only one other copied sentence (also in Methods, listed in my comments below); however, if this manuscript is to be reconsidered, the authors must thoroughly check the entire manuscript to ensure all sentences are original. Reviewer #2 has typed many comments directly on the PDF manuscript and those points need to be considered for any re submission. Finally, I have the following specific comments:

L30 Because seed predators killed some seeds, by definition the seed predators have influenced seed survival. This sentence needs careful wording for accuracy.

L39-40 Please reword to increase precision of this opening sentence. It is not clear why / how seed bank dynamics would be used to determine “when control measures would be most effective”. Do you mean for example that if germination is mainly in Spring then herbicide should be applied in Spring to kill germinating seedlings?

L143 This method of site selection is problematic in terms of reaching general conclusions about the success or failure of biocontrol agents. The agents were already released a long time ago, so any effects they have had on the Acacia populations would have been initiated in the past on populations that therefore might no longer exist or are no longer dense populations, or perhaps effects of agents have been stronger in sparse populations. Yet, you have deliberately chosen remaining monocultures of Acacia as your study sites. It is typical and fully expected that biocontrol agents will not have uniform impact across an entire region, therefore, the fact that there are remaining dense Acacia stands does not necessarily indicate failure of biocontrol agents across the region. In order to assess success or failure, we nee to identify a random set of Acacia sites from across the release region before any release has taken place and then monitor these sites over time. Given that the present study did not do this, the most that can be said is that the biocontrol agents have not been effective at the selected monoculture study sites (that conclusion seems obvious even without any study) . I think this issue requires careful consideration in the Discussion and conclusions (including the Abstract) need to be carefully worded to acknowledge limitations of the study design in terms of assessing general effectiveness of seed-killing biocontrol agents on Acacia across South Africa.

L173 These seem to be really small trees. Were the chosen sites recently cleared?

L182 The method of selecting pods for assessment is very important to ensure an unbiased sample. Methodology for pod selection should be given. Without precautions to avoid bias, damaged pods could be under-represented, leading to underestimation of attack rates. Do pods abort in response to damage? Can pod abortion rates be estimated?
L253 This sentence is copied word-for-word from Strydom et al., 2017.
L293 It would be useful to tell us the historic time frame for these biocontrol releases on Acacia. 50-100 years ago? 20-50 years ago?

L297-305 I don’t understand the reasoning in this section. You have only sampled at two sequential years (for some species) and there is no way to create a chronosequence of attack rates over time using different tree sizes. Your limited temporal sampling also does not allow for estimating an averages attack rate or seed production rate incorporating variation in disturbance (e.g. fire). I suggest deleting this entire section; otherwise it needs rewording to make a clear and convincing argument.

L325 a 95% confidence limit for what?
L330 What do you mean by “95 % model confidence set”?
L336 What gradient? There is no mention in the Methods of sampling along a gradient.
L357-364 I’m not convinced that this paragraph is needed.
L367 It would have been valuable to list the species here if you identified them.
L370-383 I would have appreciated some key summary statistics about % predated / aborted
L403 I don’t understand this phrase – can you simplify wording?
L404-417 Since no statistical tests were provided to corroborate these putative differences, I don’t think these statements about differences between years are warranted in the Results section. If you think these differences are important and worth reporting in Results, they should be tested statistically. If this is not possible, this section should be greatly condensed and/or transferred to Discussion.
L418-428 I am guesting there is a question of whether some aborted seeds are due to early predation. Anyway I think this paragraph could convey key idea(s) both more clearly and more concisely. It reads more like Discussion than Results.

L430-431 This was not a study on population dynamics and management techniques. Can this opening sentence be revised to more clearly represent the study at hand?
L432-435 I don’t see how any of these statements are clearly supported by your data. If you want to make these assertions in the Discussion, then each assertion should be immediately followed by reference to relevant data from your study. This study did not look at different degrees of success versus failure, so no clear inferences can be made about what control methods, control times, or traits are related to success or failure. My suggestion would be to organize discussion around whether the data support (or fail to support) for hypotheses. For example, hypotheses / predictions would be based on VanKlinken et al (2003) and perhaps Crawley (1989).
L439 Wouldn’t you need to know the initial seedling survival rate / emergence success rate in order to determine this?
L441-442 Is this inference too broad? – see comments on line 143 above
L454 Wouldn’t this be true for any type of plant that produces viable seeds?
L467 Would it be more accurate to say “may not have been representative”?
L478 Give the full species name and tell use what kind of organism it is.

L479 This wording is incorrect. Obviously, if no seeds had not been consumed by predation then there would have been more surviving seeds.
L501 This is the first time the genus is mentioned, so tell us what type of organism it is.
L529 Which data?
L522 Is this inference too broad? – see comments on line 143 above
L528 This seems true from your current deliberately selected monoculture sites, but is it true at other types of sites or was it always true historically?
L542 I think this should be tempered based on the data presented “at our Acacia monoculture study sites, Acacia appear not to be seed limited...”
L545 I think it is not correct English to say that seed survival is large.
L547 See comments on line 143 above. It might be true but the data presented in this manuscript seem inadequate for such a broad and confident generalization.
L553 Although the largest trees may have substantially lower seed production? See figure 1.
L555 I think this should say “management of new populations should be done during the seedling stage...” If it is an established population with adult plants then other approaches may be more efficient. Adding data on survival rates of different size / age classes would allow sensitivity analysis to identify which management actions would likely have the greatest impact on population growth rates of well-established populations such as those studied here.

Figure 1 It seems odd to me that for all three species, the largest trees had unexpectedly low seed survival. Does this indicate evidence of senescence in the larger trees?|
Table 1 – Caption “maximal models” is not clear. Expand the caption to prove necessary details.
Table 5 – Caption – I don’t understand the phrase “95% confidence set”. Please rephrase or explain. Also, if you are going to specify that “ Model averaging was conducted over the candidate models” then also tell us the purpose of doing that.
Table 6 Should the caption say “viable seeds per tree” or does it also included aborted and other non-viable seeds?
Table 7 Why do three variable have a relative importance value of exactly 1?
Table 8 – Caption – see comments on table 5

Reviewer 1 ·

Basic reporting

See my general comment

Experimental design

Hard to follow:
I find the manuscript hard to follow especially the methods. Perhaps this is a bit hard to change given the nature of the experiment, but I think the authors should improve the readability and make details more accessible to the readers to the extent possible in the revision.

Validity of the findings

See my general comment

Additional comments

The paper is not well written and has potential conflict with the Author's previous publication. (See Strydom, Matthys, et al. "Invasive Australian Acacia seed banks: Size and relationship with stem diameter in the presence of gall-forming biological control agents." PloS one 12.8 (2017): e0181763.)

I had invested time to review the manuscript and had piled a good number of issues regarding difficulty in understanding the methods in relation to the postulated hypothesis.

Unfortunately, I stopped reviewing the manuscript after reading a paragraph that is potentially plagiarized from the 2017 Plos One paper).
Line 291 to 305 in this manuscript was copied and pasted from page 3 of Strydom et al., 2017.

Similar issues can be seen in other parts of the manuscript.

This being said, I decided to keep my comments on other sections.

Reviewer 2 ·

Basic reporting

There are some grammatical issues and also redundant or extraneous material in places that need to be cleaned up but overall the paper is well written and structured.

Experimental design

Design is sufficient though there is scope for additional data analyses. These are noted in my attached comments.

Validity of the findings

Findings are valid though introduction could be edited to support the conclusions better.

Additional comments

I enjoyed your paper and commend you on the detailed work that was undertaken. I have provided comments on the attached PDF. I do think there is scope for additional statistical analyses and also the need for more information in the introduction on acacia seed ecology/ ecology. Also I think you can highlight the importance of vital rates (through the lens of reproduction and seed bank dynamics) on pop dynamics in the introduction and tie this to biological control.

Annotated reviews are not available for download in order to protect the identity of reviewers who chose to remain anonymous.

---

## Round 0.2 · accepted · Accept

The reviewer has made some comments on the PDF; please check them. Also, please define units on the x-axis of Figure 1 and Figure 2

# Reviewer 2 ·

Basic reporting

No Comment

Experimental design

No Comment

Validity of the findings

No Comment

Additional comments

Congratulations on your paper. I think it is a great piece of research. I have made some very minor suggestions (see attached PDF) to address in the proofs stage.

Annotated reviews are not available for download in order to protect the identity of reviewers who chose to remain anonymous.